# Arterial Hypertension and the Hidden Disease of the Eye: Diagnostic Tools and Therapeutic Strategies

**DOI:** 10.3390/nu14112200

**Published:** 2022-05-25

**Authors:** Rita Del Pinto, Giuseppe Mulè, Maria Vadalà, Caterina Carollo, Santina Cottone, Claudia Agabiti Rosei, Carolina De Ciuceis, Damiano Rizzoni, Claudio Ferri, Maria Lorenza Muiesan

**Affiliations:** 1Internal Medicine and Nephrology Unit, Department of Clinical Medicine, Public Health, Life and Environmental Sciences, San Salvatore Hospital, University of L’Aquila, 67100 L’Aquila, AQ, Italy; rita.delpinto@univaq.it; 2Unit of Nephrology and Hypertension, Dipartimento PROMISE (Department of Health Promotion, Mother and Child Care, Internal Medicine and Medical Specialties), Università di Palermo, 90133 Palermo, PA, Italy; giuseppe.mule@unipa.it (G.M.); caterina.carollo@unipa.it (C.C.); santina.cottone@unipa.it (S.C.); 3Biomedicine, Neuroscience and Advanced Diagnostic Department, IEMEST, Euro-Mediterranean Institute of Science and Technology, University of Palermo, 90133 Palermo, PA, Italy; maria.vadala@unipa.it; 4Department of Clinical and Experimental Sciences, Clinica Medica, University of Brescia, 25121 Brescia, BS, Italy; claudia.agabitirosei@unibs.it (C.A.R.); carolina.deciuceis@unibs.it (C.D.C.); damiano.rizzoni@unibs.it (D.R.); marialorenza.muiesan@unibs.it (M.L.M.); 5Division of Medicine, Department of Clinical and Experimental Sciences, Spedali Civili di Brescia, 25121 Montichiari, BS, Italy

**Keywords:** hypertension, microcirculation, fundus oculi, optical imaging, tomography, optical coherence, scanning laser polarimetry, coronary circulation, chronic kidney disease, antioxidants, vitamins

## Abstract

Hypertension is a major cardiovascular risk factor that is responsible for a heavy burden of morbidity and mortality worldwide. A critical aspect of cardiovascular risk estimation in hypertensive patients depends on the assessment of hypertension-mediated organ damage (HMOD), namely the generalized structural and functional changes in major organs induced by persistently elevated blood pressure values. The vasculature of the eye shares several common structural, functional, and embryological features with that of the heart, brain, and kidney. Since retinal microcirculation offers the unique advantage of being directly accessible to non-invasive and relatively simple investigation tools, there has been considerable interest in the development and modernization of techniques that allow the assessment of the retinal vessels’ structural and functional features in health and disease. With the advent of artificial intelligence and the application of sophisticated physics technologies to human sciences, consistent steps forward have been made in the study of the ocular fundus as a privileged site for diagnostic and prognostic assessment of diverse disease conditions. In this narrative review, we will recapitulate the main ocular imaging techniques that are currently relevant from a clinical and/or research standpoint, with reference to their pathophysiological basis and their possible diagnostic and prognostic relevance. A possible non pharmacological approach to prevent the onset and progression of retinopathy in the presence of hypertension and related cardiovascular risk factors and diseases will also be discussed.

## 1. Introduction

Despite major diagnostic and therapeutic advances over the past few decades, arterial hypertension still remains a paramount cause of death and a leading cardiovascular (CV) risk factor worldwide, contributing to several noncommunicable diseases [1]. With nearly 900 million adults having a systolic blood pressure (BP) ≥140 mmHg in 2015, it is estimated that one-third (31.1%) of the adult population globally has hypertension, and its prevalence has been increasing [2].

In the long-term, hypertension leads to structural and functional damage of blood vessels [3]. Such adaptive changes affect the small arteries as well as the microcirculation, and they predict the onset of hypertensive mediated organ damage (HMOD) at the level of the heart, kidney, brain, eye, and the peripheral vasculature [4] and the incidence of CV events [5]. Thanks to the availability of several screening tests and imaging modalities, the identification of HMOD in the absence of any clinical symptom is a common finding, with the advantage of allowing timely management strategies aimed at reverting or delaying the progression towards overt disease and preventing future clinical events [6]. The diagnosis of HMOD is part of the stratification of CV risk [6] and helps in the reclassification of patients towards the appropriate risk category, thereby identifying individuals who require a more intense treatment due to their increased CV risk. Successful treatment of hypertension can therefore significantly reduce the burden of incident CV diseases (CVDs) and that of hypertension-related complications [6].

At the eye level, hypertension-mediated damage hides in the most posterior part of the organ, the retina. However, in comparison with other microvascular districts, the retinal microcirculation has the unique feature of being directly and relatively easily investigated using simple tools that uncover the local presence of HMOD [7,8]. The retinal microcirculation is very similar, in terms of embryological origin, anatomic features, and physiopathology, to the far less accessible vasculature of other sites, like the heart [8], brain [9], and kidney [10]. In addition, when exposed to cardiovascular risk factors, like hypertension, the phenomenon of vascular remodeling, typically leading to an increase in the media thickness at the expense of the lumen diameter, occurs systemically [11]. Thus, for their privileged, accessible location, allowing the detection of hypertension-mediated vascular damage, the retinal microvessels have been proposed as a window to the health status of the heart [8], kidney [12], and brain [13] in the hypertensive patient.

Herein, the main ocular imaging techniques that are currently relevant from a clinical and/or research standpoint will be reviewed, together with the pathophysiological basis of their applications. The evidence behind the potential to infer the health status of kidney and coronary microvessels based on such techniques in different conditions, including diabetes, will be examined. Lastly, a possible non-pharmacological approach to prevent the onset and progression of retinopathy in the examined settings will be discussed.

## 2. Retinal Imaging Techniques

Ever since Hermann Helmholtz discovered the direct ophthalmoscope in the mid-1800s, the visualization of the retina has represented the most easily accessible way to directly and noninvasively visualize systemic microcirculation [14]. However, the clinical interest in retinal microvascular abnormalities has declined with time, due to a combination of reasons.

First, the early diagnosis of hypertension and the widespread use of antihypertensive medications made advanced retinopathy [15] a relatively uncommon finding, and the initial fundus alterations in hypertensive individuals have a controversial prognostic significance [16,17,18,19]. Second, the identification of mild retinopathy might be challenging, and the probability of an observer error very high, also due to the effect of posture, BP level, cardiac cycle, and autonomic nervous system activity on retinal vessel size [18]. Last, the Keith–Wagner–Barker or Wong–Mitchell systems for the grading of hypertensive retinopathy are based on subjective assessment by ophthalmoscopy or by fundus photographs of retinal features [18], which allow visualization of only the more superficial and larger arterioles and venules (100–300 μm in diameter) visible in the inner layers of the retinal circulation. Owing to their scarce resolution, these methods are not informative regarding the capillary network structure [20]. With their diameter of 3.5–6 μm, capillaries are indeed particularly representative of the finest vascular architecture and an early, sensitive marker of ischemia/hypoxia, well before arteriolar and venular alterations become visible.

However, about 20 years ago, Wong and collaborators [21] shed new light on the relevance of ocular fundus exploration, thanks to the introduction of a user-friendly, non-mydriatic video camera with dedicated software (inhouse image processing software, University of New Mexico, Albuquerque, USA) that allowed the recordings and off-line analysis of the retinal microvascular network in great detail. This and other subsequent computer-based imaging methodologies have the advantages of being reproducible and allowing repeated use in the same individuals for follow-up studies as well as quantitative and more standardized studies on microvascular structure and function. Measuring retinal arteriolar and venular diameters based on digital photographs also provides accurate and objective estimates, especially when arteriolar and venular widths from vessels close to the optic disc are used. Moreover, computer algorithms allow the distinction of venules from arterioles and the calculation of the so-called central retinal arteriolar equivalent (CRAE), obtained as the average of the six largest arterioles, and central retinal venular equivalent (CRVE), which is the average of the six largest venules. CRAE and CRVE, in turn, provide the arteriole-to-venule ratio (AVR). All these three measures have been associated with CV risk factors and stroke.

Thanks to more recent improvements in the field of optical imaging techniques, high-resolution, non-invasive capillary perfusion maps have been generated that allow more advanced investigations and fractal analysis of the retina [22]. The latter has been proposed as a measure of complexity of the retinal network, with evidence of some prognostic relevance in a large cohort of patients with coronary heart disease even after accounting for age, BP, and other relevant risk factors [23]. In fact, individuals with suboptimal retinal complexity, namely those in the lowest and highest quartiles of retinal vascular fractal dimensions, experienced increased mortality for coronary heart disease as compared with those with optimal fractal dimensions after 14 years of follow-up [23]. Moreover, advances in retinal vascular imaging have allowed us to gain important insights into the branching pattern of the retinal microvascular network [16,17,20,21]. Retinal photographs allow the definition of novel indices of the retinal vascular network architecture, e.g., changes of curvature tortuosity and of branching angle, thereby enabling the identification of suboptimal vascular branching patterns suggestive of microvascular damage in the cerebrovascular, coronary, and renal districts [24,25].

Notably, images obtained with the use of these technologies are not informative regarding the vessel wall. Progress from this point of view was provided by Harazny, Michelson and Schmieder, who in 2007 proposed a new, non-invasive methodology for the study of the retinal vascular district based on the assessment of the wall-to-lumen ratio (WLR) of retinal arterioles by means of scanning laser Doppler flowmetry (SLDF) (Heidelberg Retina Flowmeter, Heidelberg Engineering, Heidelberg, Germany) and dedicated software analysis (SLDF version 3.7 by Welzenbach) [26,27]. This approach combines a confocal measurement of the external diameter of retinal arterioles (reflection images) with that of their internal caliber using a laser Doppler technique (perfusion images) [26,27]. The two images taken in the same retinal area are automatically compared using dedicated software [28]. Using this approach, the WLR of retinal arterioles was found to be prognostically relevant, being particularly elevated in individuals with hypertension and cerebrovascular disease [29] and directly correlated with urinary albumin excretion [30]. SLDF is also able to detect the pulsatile characteristics of the retinal arterioles, a parameter that was found to be differentially impaired in patients with low–medium grade hypertension and in those with treatment-resistant hypertension [31]. In addition, based on two-dimensional mapping of perfusion variables, SLDF allows the visualization of perfused retinal vessels and the optic nerve head [32]. Retinal flow and arterial dimensions might be measured in basal conditions or after specific stimuli, such as a flicker light. A luminance flicker light with a frequency of 8 Hz was shown to increase retinal capillary blood flow [33] as well as blood flow in retinal arteries and veins [34] in healthy individuals. The flicker-induced dilatation of retinal arterioles was demonstrated to have some prognostic relevance, since it is reduced in patients with CV risk factor and, to a greater extent, in patients with heart failure compared with healthy controls [35]. Some preliminary evidence suggests an increase in the WLR of retinal vessels when measured with SLDF in high-risk populations [26,36,37]. SLDF has several strengths, including easy repeatability with minimal discomfort for patients, a less than 10% interobserver and intraobserver variability in the estimate of the WLR [28], and the possibility to gather functional information. However, the reproducibility of WLR measurement in real life settings may be affected by pitfalls in the correct estimation of the internal diameter based on the Doppler technique [36]. Another major limitation is the fact that the Heidelberg Retina Flowmeter is not substantially commercialized, thus limiting the clinical development of this approach [7].

Some years ago, a novel technique became available for the direct measurement of the WLR of retinal arterioles based on the adaptive optics (AO) imaging system (AO camera, Rtx-1, Imagine Eyes, Orsay, France) [37,38]. Originally used to correct for aberrations in astronomical optical systems [37], this technique allows for the investigation of vessels whose diameter ranges from 20 µm to over 150 µm [39]. The principle of AO is the partial reflection into the optical system of a beam of light directed into the eye [37]. A dedicated image sensor in the system senses the wavefront aberrations in the reflected image, which are then corrected by a deformable mirror [39,40].

The system provides images with unprecedented quality and resolution, with a resolution of the order of 1 µm [7] if the eye fixation is correct and the ocular media are transparent [33]. AO provides a substantially stable estimation of the morphological parameters of retinal arterioles for a length of the examined segment of at least 500 µm [37]. WLR evaluated by AO was demonstrated to be linearly related with age [41,42] and with BP values [37], and it was reduced only in effectively treated hypertensive patients, while no change was observed in patients whose BP did not decrease during treatment [38]. The performance of AO for the assessment of the WLR of retinal arterioles was demonstrated to be superior to SLDF, being more reproducible and more closely correlated with the media-to-lumen ratio (MLR) of subcutaneous small resistance arteries [36]. Nevertheless, some limitations exist that prevent the widespread application of AO in the clinical setting, spanning elevated costs to time issues as well as the lack of normative databases, all of which add to an intrinsic limitation of the technology in terms of the limited size of the examined area. However, it remains a remarkable research tool that enables the visualization of the retina at a cellular level [40].

Among the new vascular imaging techniques in ophthalmology, by far the most used in clinical practice is optical coherence tomography (OCT), which is chiefly used for the diagnosis of sight-threatening retinal disease, but which can also be employed for the detection of ocular involvement during systemic disease and to monitor its progression during treatment [42]. OCT is a light-based, cross-sectional imaging modality that does not require contact with the eye. In analogy with ultrasonography, it examines the reflections of a probe beam directed onto a biological tissue to obtain depth information. Specifically, this technique directs a broadband light source to the retina, and the interference pattern of the backscattered waves with the reference beam is used to generate high-resolution cross-sectional images of the retina, its nervous structure, and the optic nerve head [42,43]. The time delay of the returning light reflection—a measure of tissue depth—is determined indirectly by low-coherence interferometry, where the light returning from the tissue and that from a reference arm are combined in an interferogram. More modern variants of OCT have also been developed, including the spectral domain (SD) and the swept-source (SS) OCT, which use microelectromechanical systems to generate lights with high performance. SD-OCT is based on the use of a broad-spectrum light source, and the reflectivity of the tissue impacts on the resulting interferogram. In SS-OCT, a more complex light source sequentially scans through successive wavelengths of light across a defined spectral range, leading to the generation of as many interferograms beams of light. The advancement in modern OCT technology and the use of enhanced depth imaging (SD-OCT and SS-OCT) have led to the high resolution visualization of specific cell layers within the retina, as well as that of deeper structures like the choroid [42,43]. This achievement, in particular, is a strength of OCT technology, since the study of this thin, dense vascular layer sandwiched between the retina and the sclera has always represented a challenge in the use of conventional imaging modalities, such as indocyanine green angiography and ultrasonography, or with methods based on partial coherence interferometry, due to limited image resolution and repeatability [41,44]. The introduction of SD-OCT has enabled us to capture cross-sectional images of the choroid in living eyes, thereby greatly increasing our understanding of this tissue [42,43].

However, SD-OCT suffers from several inherent limitations. First, the detection of the choroidal-scleral boundary is not always accurate, as it requires the averaging of 50–100 B scans to achieve high-contrast and low-speckle noise, and it might suffer from low penetration through the retinal pigment epithelium (RPE) [42,43]. In addition, the choroid may present focal thickening or thinning, which may affect the accuracy of manual segmentation obtained at a few points and applied in most of the studies using SD-OCT, resulting in high variability of the results. Vice versa, SS-OCT shows several advantages compared to SD-OCT in the study of the choroid as well in measuring choroid thickness, since it employs a wavelength-tunable laser as a light source and operates through a dual-balanced photodetector. Furthermore, it may adapt to longer imaging wavelengths, and it works with higher speed, which makes the image acquisition and three-dimensional image reconstruction more accurate [42,43].

Other vascular parameters with a clinical importance that can be measured in the retina may be obtained with recent techniques, such as Laser-speckle contrast imaging (LSCI) or optical coherence tomography angiography (OCTA) [18,29,45]. LSCI is a real-time, full-field, non-invasive technique that enables the study of changes in the retinal blood flow [29] though it also may be applied in other sites [46]. It is grounded in speckle contrast and the dynamics of erythrocytes that behave as light-scattering particles [46] and provides information in terms of modifications in the perfusion patterns. However, its use in clinical settings at present is limited because of some concerns in the underlying algorithms [46]. OCTA is another new non-invasive technique based on blood flow imaging for a three-dimensional representation of the retinal flow, which enables a reproducible study of retinal microvessels by analyzing the erythrocyte movement without the need for contrast medium [47]. Through the analysis of repeated scans of the erythrocyte movements, which are in contrast with static elements such as retinal cells, this technique may represent a useful tool to assess structural alterations in the retinal microvessels of patients with hypertension [48]. Moreover, OCTA is equipped with software (software ReVue version 2014.2.0.93, Optovue Inc., Fremont, CA, USA) for the automatic quantitative evaluation of several parameters of the retinal vascular network, such as vascular density, defined as the percentage of area occupied by blood vessels. Vascular density is usually calculated for two different regions of interest (ROI), which include the foveal and parafoveal areas. Using OCTA, it was demonstrated that hypertensive patients show a reduction of retinal vessel density as compared to control individuals at several retinal sites, such as the fovea and parafovea. OCTA could theoretically represent a potential technique for the evaluation of retinal vessels and the follow-up of patients with hypertension [49], although limited clinical data about its clinical application are available at present.

## 3. Retinal and Glomerular Vasculature: Two Sides of the Same Coin

Arterial vessels in the retina share anatomical and physiological features with the same vessels in the kidney, brain, and heart, suggesting that all might suffer from a similar disease burden when exposed to risk factors and conditions affecting the vascular system, and that changes occurring in the retinal vessels may be the mirror of similar damage in other tissues. It has long been known that the severity of ocular fundus abnormalities predicts mortality in hypertensive patients [15]. Changes in retinal vascular diameter have also been independently associated with the incidence of stroke [15], CV diseases and mortality [24], and kidney failure [25]. Consistent with this, meta-analytic evidence indicates that the narrowing of retinal arteries and widening of venulae have an independent association with increased hypertension risk [50]. Moreover, in nondiabetic hypertensive individuals, the rarefaction of retinal microcirculation as assessed by OCTA was associated with impaired renal function [47]. Indeed, lower values of retinal vascular density were observed in individuals with eGFR < 60 mL/min/1.73 m^2^ when compared to those with eGFR ≥ 60 mL/min/1.73 m^2^, independent of confounders (Figure 1). Thus, such retinal changes may represent useful and easily accessible markers of hypertension-related end-organ damage, thereby adding a paramount contribution to the stratification of CV risk in hypertensive patients. Moreover, recent investigations using OCTA technology suggest retinal capillary rarefaction in individuals with diabetes [45,51] and in those with arterial hypertension [52]. In both conditions, the greatest capillary rarefaction was generally observed in individuals with the lowest eGFR.

The choroid is a pure vascular structure, with the highest blood supply in comparison with other body sites, which is critical for the perfusion of the external one-third of the retina, the RPE, the foveal avascular zone (FAV), and the optic nerve [49,50]. It exerts this critical task under relatively high perfusion pressure; thus, subtle changes in the latter may affect retinal oxygenation and functioning [53]. The choroid participates in the pathophysiology of several retinal diseases and can be directly damaged by acute increases in BP as well as by severe, long-lasting hypertension [44].

There is an analogy in terms of anatomy and physiology between the choroidal circulation and the glomerular vascular network. With its large surface area and interdigitating foot processes, the renal podocyte and the vascular pericyte are very similar in structure and function [54]. The glomerular filtration barrier and the retinal barrier share similarities as well. With its ~80 nm fenestrations, the choriocapillaris endothelium allows the exchange of fluid within the subretinal space. The glomerular endothelium has fenestrations of similar size that contribute to ultrafiltration into the Bowman’s capsule [54]. Responsiveness to the renin-angiotensin-aldosterone system has also been found in the retina, where it targets several retinal components, including ganglion cells, Müller cells, and RPE [54]. Angiotensin II induces endothelial dysfunction and elicits inflammatory responses mediated by an increased production of oxygen free radicals. Other mechanisms of tissue damage induced by angiotensin II include matrix remodeling, gene expression regulation, and the inactivation of several intracellular signal transduction pathways. Hence, it is possible that reduced retinal vascular density or established retinal or choroidal lesions reflect similar changes in the kidney of hypertensive patients.

Growing evidence suggests that changes in choroidal thickness (ChT) represent proxy measures of segmental altered perfusion, possibly underlying a more generalized vascular damage [55,56]. However, studies exploring ChT in patients with CV risk factors or with CVD yielded inconsistent results. Isolated hypercholesterolemia has been linked to ChT [57], whereas tobacco smoking [58], ocular ischemic syndrome [59], chronic heart failure [60], and coronary artery disease [61] have been associated with a thinner choroid. Variable associations with ChT were reported for diabetes without retinal damage [62,63] as well as for carotid artery stenosis [64,65]. Choroidal thinning may also depend on an increased sympathetic tone of choroidal vasculature, since, unlike the retina, the choroidal circulation receives an autonomous innervation. CKD, which is characterized by a sympathetic nervous system hyperactivity, is theoretically a condition associated with a thinner choroid [66].

Data on the relationships between mild to moderate essential hypertension and hypertension-mediated renal damage with ChT are sparse and conflicting [67,68,69]. While some authors [70] found no relationship between systolic (SBP) and diastolic BP (DBP) with ChT, and no difference in ChT between hypertensive subjects and normotensive controls, others [67] found choroidal thinning in the presence of hypertension but not in the absence of this condition. The effect of renal function on the relationship between choroidal thickness and BP was not assessed in this study. Moreover, a study on 116 hypertensive patients undergoing 24 h ABPM and SD-OCT did not confirm the presence of any significant relationship of ChT with 24 h SBP and DBP values [71], in agreement with previous reports [68,69]. However, more recent evidence from 155 hypertensive subjects indicates a significant inverse correlation of ChT with 24 h brachial pulse pressure (PP), and even more so with 24 h estimated aortic PP [68,69].

In a group of 100 patients with essential hypertension, individuals with subclinical renal damage (defined as eGFR 30–60 mL/min × 1.73 m^2^ or albuminuria > 20 μg/min) showed increased ChT, as assessed by SS-OCT, compared with hypertensive patients with normal renal function [68] (Figure 2). The influence of renal function on ChT was confirmed in a wider population of hypertensive non-diabetic individuals, including patients with renal disease unrelated to high BP, where albuminuria and eGFR were found to be inversely and directly related to ChT, respectively [69]. Similar results come from a total of 1395 ocular treatment-naïve Chinese patients with type 2 diabetes, where ChT decreased in parallel with renal impairment [72]. Moreover, in a population of 150 individuals (50 with hypertension, 50 with CKD, and 50 healthy controls), a direct association between eGFR and ChT was reported, with evidence that a thinner choroid was associated with increased serum levels of well-known biochemical markers of oxidative stress, inflammation, and endothelial dysfunction, namely asymmetric dimethylarginine, C-reactive protein, IL-6, and endothelin-1 [70].

Notably, a close association was described between reduced ChT and increased intrarenal resistance index (RRI), which is a marker of impaired intrarenal hemodynamics as well as of extrarenal organ damage [73,74,75], being closely associated with carotid intima-media thickness [73] and with aortic stiffness [75]. It is well established that large arteries have elastic properties that ensure the dampening of pressure and flow fluctuations generated at every ventricular contraction, in order to protect the microvascular level from elevated PP and from damage due to hyperperfusion. However, arterial stiffening that occurs with aging or hypertension and other disease states prevents such adaptations and exposes the microvessels to increased pulsatile stress, with particularly relevant consequences in high-flow/low-resistance organs like the brain and kidney, whose capillary networks may be particularly sensitive to excess pressure and flow pulsatility.

Similar to what happens in the kidneys, arterial stiffness might predispose choroidal circulation to increased hemodynamic pressure, with consequent local damage and reduced ChT. In line with this concept, significant inverse relationships of ChT with PWV [70] as well as with aortic PP [76] have been reported.

To date [6], it is recommended that HMOD in multiple organs, or multiple markers of a single organ impairment (as it is the case of eGFR and albuminuria as markers of kidney function) are actively searched in individuals with hypertension, because CV risk increases with the presence of multiple signs of HMOD. Future studies are required to determine if retinal capillary rarefaction and/or reduced ChT can represent an early and accurate marker of hypertensive damage, including that at the kidney level.

## 4. Retinal and Coronary Microcirculation

Coronary microcirculation shares several characteristics with cerebral and retinal microcirculation, including the same anatomical and embryological origin (ectodermal) and physiological and pathophysiological features. In several systemic vascular conditions and diseases, including arterial hypertension, atherosclerosis, and coronary artery disease, an association was described between retinal arteriolar alterations (microaneurysms or exudates) and systemic vascular damage. Some reports indicate that structural alterations of the retina are an early marker of concomitant coronary artery disease and of its severity [77,78]. Microvascular angina (MA) is a clinical condition characterized by myocardial ischemia without evidence of obstructive coronary artery disease [79]. Patients with MA may frequently have microvascular dysfunction, with an associated increased risk of major cardiac events, including myocardial infarction (MI), heart failure with preserved ejection fraction, and stroke [80]. Furthermore, in the Multi-Ethnic Study of Atherosclerosis (MESA study), a relation of retinal changes with coronary artery calcification and myocardial perfusion was described [81,82].

Coronary microvascular dysfunction (CMD) may result from functional or structural alterations, or both, at the level of the coronary microvessels. Specifically, functional abnormalities may typically consist of an impaired vasodilator response and/or microvascular spasms leading to vascular constriction of coronary microvessels [80]. The impairment of vasodilation may depend on endothelium-dependent and/or endothelium-independent phenomena. Several traditional CVD risk factors, including hypertension, diabetes mellitus, and dyslipidemia, were shown to induce endothelial dysfunction, with consequent vasoconstriction and frank reduction in coronary blood flow [80]. The increase in substances with a vasoconstrictor effect, as well as enhanced sympathetic activity, have also been proposed as potential mechanisms for reduced vasodilation. Reactive oxygen species and inflammation may then represent the final common pathways leading to microvascular dysfunction in individuals exposed to CV risk factors [80]. Microcirculation response to functional stimuli may be also influenced by heart rate, the duration of diastole, and left ventricular contractility [80].

The inward remodeling of coronary arterioles with an increase in WLR and myocardial capillary rarefaction are the abnormalities that characterize structural remodeling of coronary microcirculation [83]. The increase in minimal myocardial vascular resistance leads to an impairment in blood flow and oxygen supply to the myocardium during stress or exercise [79,83]. CV risk factors (cigarette smoking, hyperlipidemia, high BP, insulin resistant states, and overt diabetes) may cause microcirculation remodeling, alone or in the presence of obstructive atherosclerosis or cardiac remodeling, namely left ventricular hypertrophy [83].

In the absence of obstructive coronary stenosis, MA is diagnosed when at least one of the following conditions are detected: myocardial ischaemia on non-invasive testing; reproducibility of symptoms in association with ECG changes, in the absence of epicardial spasm during coronary function test [79]. Non-invasive approaches recently proposed for the diagnosis of MA include Transthoracic Doppler Echocardiography (TTDE) with measurements of the maximal diastolic flow at the level of epicardial arteries (left anterior descending artery) during rest, and adenosine/dipyridamole/regadenoson stress with the assessment of the ratio between the peak flow during infusion and the basal flow, known as coronary flow velocity ratio (CFVR) [79]. In addition, Single Photon Emission Computed Tomography (SPECT) and cardiac Positron Emission Tomography (PET) may identify myocardial perfusion defects suggestive of coronary microvascular abnormalities [79]. Myocardial regional and global perfusion, at rest and during stress, as well as coronary resistance and flow reserve, may be obtained by these techniques [84].

Some evidence indicates that, in hypertensive patients, there is a significant correlation between the remodeling of subcutaneous small resistance arteries and coronary flow reserve, as assessed by transthoracic Doppler echocardiography with measurements of the maximal epicardial diastolic flow during rest and stress, as described above [83]. On the other hand, the same alterations are strictly related to WLR of retinal arterioles [36], making the retina a sort of “window” to the heart health status [8]. All the CV risk factors that favor the development of functional and structural alterations of coronary microcirculation have been demonstrated to affect other microcirculatory districts, including the retinal vessels. In agreement with this are the results of a preliminary study on 19 individuals suffering from angina in the absence of obstructive coronary artery disease at the coronary angiography and SPECT stress hypoperfusion (11 males, mean age 71 + 6 years) and 18 healthy controls with normal SPECT response (10 males, mean age 69 ± 6 years), who were assessed for retinal arteriole morphology [7]. WLR of retinal arterioles was measured in all participants by means of an AO imaging system (Imagine Eyes, Orsay, France), while BP was measured using an automated oscillometric device (Omron HEM 9000Ai, mean of three measurements). No differences in demographic and hemodynamic characteristics between groups were detected, except for a higher body mass index (BMI) in patients with MA (30 ± 4.2 kg/m^2^) compared with controls (24 ± 4 kg/m^2^) (*p* = 0.001). Mean systolic BP values were similar between groups (131 ± 16 vs. 126 ± 20 mmHg, respectively), and so was the prevalence of hypertension (90%), diabetes mellitus, smoking, and treatment rates with statins, with a trend towards a non-significant, slightly higher prevalence in patients with MA than in those without. Retinal arteriole wall thickness, WLR, and wall cross-sectional area were higher in patients with MA as compared with controls, and all differences were confirmed after adjustment for BMI (Table 1). No differences in internal and external vessel diameters were recorded. These preliminary findings suggest that structural changes in the retina may be observed in patients with MA and normal epicardial coronary arteries. However, in addition to body weight, the impact of other relevant risk factors (e.g., slightly higher BP, diabetes, smoking, and dyslipidemia) on the development of both coronary and retinal small arteries cannot be excluded.

## 5. Diabetic Retinopathy

Diabetic retinopathy is a microvascular complication of diabetes mellitus, representing a paramount cause of vision impairment and blindness worldwide, which is also due to the high co-prevalence rates of diabetes and hypertension [85].

Hyperglycemia induces retinal damage through complex metabolic pathways involving inflammation and oxidative stress, with consequent vascular damage, capillary ischemia, and retinal tissue hypoxia [85]. Persistent hypoxia results in growth factor deregulation, with an increase in vascular endothelial growth factor (VEGF) and other proangiogenic factors, thereby leading to proliferative diabetic retinopathy and the progression of visual impairment [85]. Other pathogenic mechanisms include increased renin-angiotensin systemic activation, increased vascular permeability and neural degeneration, as well as abnormalities in retinal veins [85].

Early microvascular abnormalities of diabetic retinopathy consist of thickening of the basement membranes, loss of pericytes, capillary occlusion, and the formation of microaneurysms [86]. On the other hand, neural degenerative changes occur in the retina in association with diabetes, from apoptosis of different retinal cells such as photoreceptors, bipolar cells, and ganglion cells, which are particularly sensitive to hypoxia [87]. Such neural dysfunction may develop in parallel or even before the establishment of microvascular lesions, resulting in the occurrence of changes in the vascular elements [88]. Indeed, the retina is more than a network of blood vessels; in fact, like the brain, it represents a complex neurovascular unit. In keeping with this, in 2017 the American Diabetes Association reclassified diabetic retinopathy from a microvascular complication to a neurovascular complication [89]. Neurovascular coupling seems to affect retinal metabolism and affect the caliber of retinal blood vessels though mechanisms that are still partially unknown. Some evidence suggests that pericytes, located in the retinal arterial walls but extending their processes into the perivascular tissue, might be involved in the process [90].

Clinical diagnosis of diabetic retinopathy is traditionally grounded in the detection of microangiopathy by means of ophthalmoscopy, conventional fundus photography, and intravenous fluorescein angiography. Recently, it has been proven that AO retinal imaging can be a useful tool to assess retinal arteriole parameters in vivo [37] in order to quantify early pathological changes of both cone density and retinal arterioles associated with diabetes mellitus in patients with no diabetic retinopathy [91,92,93], thus providing information with respect to the incidence and progression of the disease.

In particular, the lumen of parafoveal retinal capillaries, evaluated with an AO camera, were found to be narrower in eyes with nonproliferative diabetic retinopathy compared to healthy controls [91]. Similar results have been observed in arteriolar parameters from type I diabetic patients using the AO technique [93]. These changes result in arterial remodeling and increased WLR, which may reflect wall thickening, lumen narrowing, or both phenomena. In diabetic patients, retinal arterioles narrow their lumen because of fibrosis and the proliferation of smooth muscle cells [94]. These changes represent an early stage of diabetes-associated vascular impairments.

However, retinal arterioles were also found to abnormally dilate in the early stages of diabetic retinopathy [95], possibly due to disturbances in neurovascular coupling or retinal metabolism and excessive release of vasodilating mediators. The dilation of retinal vessels may lead to hyper-perfusion, allowing hemodynamic perturbations related to high BP to impact on the capillary bed, with consequent microvascular damage such as microaneurysms and hemorrhages. Moreover, this retinal arteriole remodeling is associated with alterations in pressure and metabolic autoregulation, with consequently reduced vessel capacity to adapt their diameter in response to changes in the arterial BP or in retinal metabolic requirements. The impaired autoregulation observed in diabetic patients could also depend on structural changes in the vascular walls attributable to accelerated arteriosclerosis.

Interestingly, a significant association between narrower retinal arteriolar calibers, obtained by computer-assisted evaluation of retinal photographs, with incidence of lower extremity amputations and all-cause and stroke-related mortalities was described in patients with type 2 diabetes [96,97,98]. Similarly, a wider venular diameter was found to be associated with incidence of nephropathy and with increased stroke-related mortality [99]. Moreover, the increased venular diameter measured by fundus photographs has been significantly correlated to the severity of diabetic retinopathy [100].

Indeed, retinal arteriolar narrowing and arteriovenous nicking are more typically associated with hypertension, while retinal venular widening could reflect retinal hyperperfusion and/or inflammation associated with hyperglycemia and hypoxia [99,101] and may subtend similar changes in the cerebral, coronary, peripheral, and renal circulations. Indeed, hypertrophic remodeling was observed in subcutaneous resistance arteries from type 2 diabetic patients compared with nondiabetics [102,103]. This was associated with a significant loss of myogenic responsiveness in diabetic patients compared with control patients [103]. A reduced dilator response to flicker stimulation has also been observed in diabetic patients compared with healthy subjects [104,105], even before diabetic retinopathy was established. Therefore, changes in the size of retinal arterioles and venules with an increased retinal arteriole WLR can be detected in diabetic individuals as an expression of microvascular disease.

## 6. Nutrition and Retinal Alterations

As mentioned, structural and functional alterations in the retinal microvessels are usually detected in cardiometabolic diseases [3], such as diabetes mellitus, hypertension, and dyslipidemia, where nutritional interventions may have a relevant role. Indeed, the impact of diet on the microcirculation was documented by studies using capillaroscopy or ocular imaging techniques to assess the effect of such interventions at the level of the skin or the retina, two easily accessible sites for the study of microvessels. For instance, a randomized controlled trial of 137 men and women aged 40 to 70 years that compared compliance to UK dietary guidelines with a traditional British diet [106] found an increased capillary recruitment in the skin, as assessed by capillaroscopy, in individuals on a diet compared with the counterpart [106]. The intervention diet was characterized by a decreased amount of total and saturated fats, simple carbohydrates, and salt, and an increased content of greens, seeds, and oily fish. Similarly, among the 2720 Blue Mountains Eye Study participants (adults aged 50 years and older), compliance with a guideline-recommended, high-quality diet (e.g., based on whole-grain cereals, low-fat milk, lean meat, reduced saturated fat intake, and increased fish consumption) was associated with better retinal microvascular health, expressed as larger retinal arterioles and narrower venules, compared with lower-quality eating habits [107]. By contrast, high-carbohydrate, high–glycemic index (high-GI) diets were associated with small vessel dysfunction even at a very young age. In fact, among 2353 adolescents aged 12 years selected from a random cluster sample of 21 schools, increased carbohydrate consumption and the intake of soft drinks was associated with retinal arteriolar narrowing and venular widening [108]. Given the prognostic relevance of these abnormalities in terms of incident CV events, their presence at such a young age demands attention and requires drastic public health interventions aimed at supporting healthy eating patterns from youth onwards [108].

It is not known whether dietary integration with specific nutrients is beneficial in terms of retinal microvascular health. In the last decades, however, there has been a growing interest in studying the protective effects of food supplements, such as polyphenols, omega-3-polyunsaturated fatty acids, and vitamins, in chronic diseases of the eye. The mechanisms behind their benefits for retinal health encompass the preservation of endothelial function, neuronal protection, antioxidant and anti-inflammatory properties, regulation of angiogenesis and apoptosis, as well as BP reduction [109,110,111,112].

Oxidative stress, i.e., the excess production of reactive oxygen species, is a major contributor to retinal damage [113,114]. It exerts a mechanistic role in photoreceptor cell death and vascular damage in several diseases affecting the retina, regardless of association with cardiometabolic risk factors like diabetes (e.g., diabetic retinopathy) or hypertension (e.g., age-related macular degeneration (AMD), a chronic disease of the eye leading to reduced vision or even blindness that is often associated with hypertension). Indeed, due to a high metabolic rate, the human retina must withstand significant levels of oxidative stress, which leads to microvascular damage [112]. Oxidative stress is induced by free radicals, as well as by nitric oxide inactivation, and causes endothelial dysfunction, increased media thickness, thrombosis, and the production of angiogenic and inflammatory factors, which concur in determining retinal damage [114].

Therefore, antioxidants could play a critical role in preserving cellular functions, particularly at the level of the retina. B vitamins, especially B2, B6, B9, and B12, represent sources of coenzymes supporting a broad set of transformations, known as one-carbon (1C) metabolism, which participate in the conversion of homocysteine (HCY) to methionine. Previous studies found that elevated serum homocysteine is associated with diabetic retinopathy and increases microvascular damage [115,116]. High serum levels of homocysteine determine a reduced production of NO that contributes to microvascular disease, namely endothelial damage and apoptosis, capillary leakage, and tissue ischemia, all of which contribute, at the level of the retina, to vascular atrophy and neovascularization and, ultimately, to vision loss [100]. Interestingly, decreasing homocysteine levels with folic acid supplementation in adults with hypertension complicated with diabetes mellitus was associated with reduced risk of retinal microangiopathy, defined in accordance with the Keith–Wagener–Barker classification system [117] and with better retinal perfusion, possibly due to an increase in retinal artery caliber [118].

Among the antioxidant compounds, a largely abundant family is that of carotenoids, natural pigments whose biochemical structure attracts electrons from reactive species and neutralizes free radicals, thereby scavenging oxidative species [119]. Lutein and zeaxanthin, two water-soluble carotenoids concentrated in the macula but not synthesized in humans, have been demonstrated to reduce the risk of chronic eye diseases, including AMD, by exhibiting neuroprotective, antioxidant, and anti-inflammatory properties [110,120]. In addition, in the Carotenoids in Age-Related Eye Disease Study 2 (CAREDS2), conducted in older women participating in the Women’s Health Initiative, higher serum levels of lutein and zeaxanthin at baseline were associated with wider central retinal vessel calibers during follow-up, which, in turn, was related to lower systolic BP and lower likelihood of diabetes, suggesting another possible mechanism behind vision protection by carotenoids that is mediated by modulation of ocular blood flow [121].

The fat soluble, strongly antioxidant α-tocopherol, the most common active form of vitamin E, is another promising therapeutic option in the management of retinal diseases. It exerts anti-apoptotic and anti-inflammatory effects involved in the protection against retinal degeneration. According to studies in rats and apes, vitamin E–deficient diets induced changes in the membrane microenvironment, leading to photoreceptor degeneration mediated by lipid peroxidation, membrane fluidity impairment, and irreversible loss of long-chain polyunsaturated fatty acids (LC-PUFAs) [122,123,124]. In AMD, the combination of a pro-inflammatory state with oxidative stress contributes to deposit formation, which triggers the activation of complements, thereby leading to disease progression, and treatment with vitamin E was found to prevent such molecular events [125]. Tocotrienol, the other active form of vitamin E, was shown to inhibit angiogenesis, which is a crucial event in AMD, and this property was at least in part due to the regulation of vascular endothelial growth factor (VEGF) signaling [126].

Similar to vitamin E, resveratrol is a polyphenol, mostly found in grapes and berries, that has well-described anti-oxidant and anti-inflammatory properties of relevance for the retinal health status [127]. In addition, it also has anti-proliferative effects, inhibits platelet activation and aggregation, and facilitates the endothelial synthesis of NO [127]. In vitro studies on the effects of resveratrol in AMD and diabetic retinopathy are promising [128,129]. In vivo experiments in octogenarians supplemented with resveratrol showed improvements in choroidal blood flow and in clinical outcomes [130]. Similarly, supplementation with resveratrol was found to decrease diabetic retinopathy progression in diabetic patients [131,132]. In analogy with the effect of resveratrol, a rich diet in omega 3 fatty acids was proven to be effective in slowing disease progression in diabetic retinopathy thanks to its antioxidant, anti-inflammatory, and anti-angiogenic properties [131]. Another compound with antioxidant properties is the amino acid taurine, which was demonstrated to prevent diabetes-induced apoptosis in retinal glial cells via its anti-oxidation and anti-glutamate excitotoxicity mechanisms [133] apoptotic [133] of high glucose levels on retinal microvascular pericytes [133].

A certain degree of antioxidant capacity has also been described for vitamin D. In fact, it was demonstrated by in vitro studies that vitamin D supplements may prevent intracellular reactive oxygen species formation, consequently reducing the expression of VEGF [112]. In addition, it has been reported that vitamin D deficiency is associated with increased risk of high BP [134] and with increased risk and severity of diabetic retinopathy [135]. There is also evidence linking vitamin D with inflammatory conditions [136,137]. However, inconsistent findings from in vivo studies limit the therapeutic applications of vitamin D supplementation in comparison to those related to bone health [138].

An aspect worth mentioning in the non pharmacological management of hypertension and blood vessel health status relates to the gut microbiome and its intertwined relation with diet [139]. In the past few years, next-generation sequencing technologies have revealed the contribution of the intestinal flora to a variety of pathways related to CV function in health and disease states, including hypertension [139]. For instance, evidence indicates that both fasting and high-fiber food promote a switch in the microbiota composition towards short-chain fatty acid (SCFA)-producing species [140,141]. SCFA, including the most abundant acetate, butyrate, and propionate, are fermentation products of indigestible fibers and promote the differentiation of immune cells towards an anti-inflammatory phenotype [142] while also exerting some antioxidant activity [143]. They have also been involved in the epigenetic regulation of several cell types. Thus, microbial and immune changes related to fasting and a high-fiber diet have been associated with decreased BP. Interestingly, it has been reported that gut dysbiosis triggers immune cells specific for the retina, and that the resulting low-grade inflammation in genetically predisposed individuals aggravates dystrophic retinal changes that contribute to AMD [144]. Hypertension contributes to the development of AMD in several ways, including impaired choroidal perfusion and local metabolic impairment responsible for regional atrophy and pathological revascularization, and the systemic neurohumoral events affecting vascular function triggered by gut dysbiosis have been also reported as possible determinants of the disease [145]. Similarly, dysbiosis induced by a high-fat diet was found to trigger proinflammatory cytokine production that worsened pathological neo-angiogenesis of the choroid in AMD [146].

A potential benefit might therefore derive from a nutraceutical approach to microvascular disease. However, very few data are available about the effects of dietary intervention and dietary supplementation on retinal vessels, and this issue deserves further evaluation in specifically designed intervention studies.

## 7. Conclusions

Non-invasive approaches for the assessment of microvascular morphology and function are urgently needed for a better risk stratification of individuals at increased CV risk, as well as for the monitoring of the protective effects of cardiovascular drugs. The possibility to measure the WLR of retinal arterioles with reliable approaches such as AO may represent a major advancement in this regard, thus making the assessment of ocular fundus an optimal resource for the non-invasive estimation of hypertension-induced changes in the microvessels. However, there is still a lack of unequivocal evidence that the WLR in retinal arterioles has some prognostic relevance [11]. Overall, the data available with such approaches are, at present, somewhat limited. In addition, while representing a desirable goal, a method for a reliable, non-invasive estimation of the endothelial function in the microcirculation is still lacking, though techniques such as the flicker-light-induced dilatation of retinal arterioles might provide some real advancement in this sense.

In conclusion, new and emerging technologies, some of which are currently under clinical testing, may possibly help in the non-invasive assessment of microvascular structural and functional alterations in the near future, thus allowing a better stratification of CV risk and consequent optimization of treatment [7]. In a time when interest in the non-invasive assessment of microvascular structure is on the rise, determining the prognostic value of related findings is mandatory. At present, the evidence in support of an extensive application in the clinical setting of such available techniques is sparse, requiring further testing in dedicated, properly sized studies in specific populations.

## Figures and Tables

**Figure 1 nutrients-14-02200-f001:**
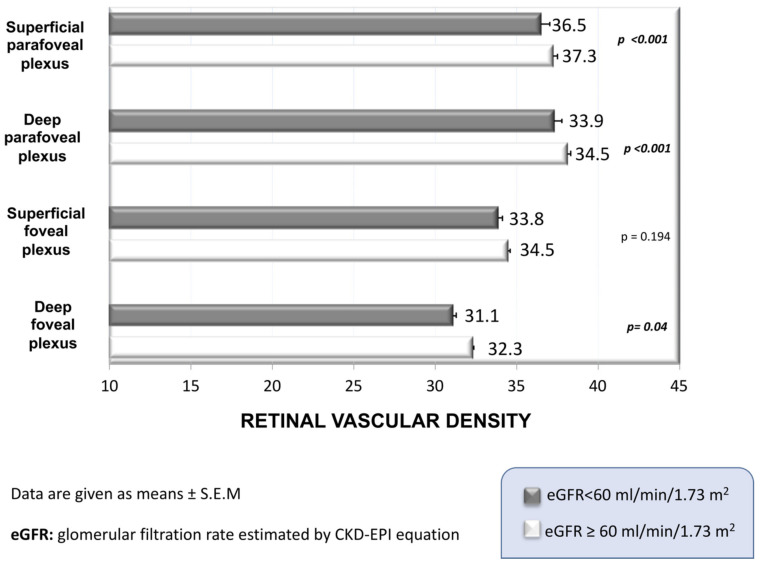
Differences in retinal vascular density between hypertensive subjects with eGFR above and below 60 mL/min/1.73 m^2^.

**Figure 2 nutrients-14-02200-f002:**
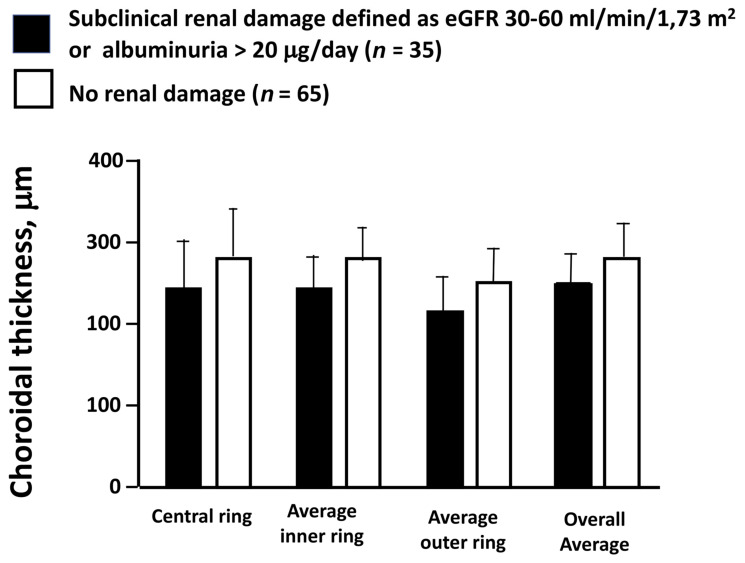
Mean values ± SD of ChT, measured as the overall average of all nine regions examined according to the Early Treatment Diabetic Retinopathy Study (ETDRS) protocol, and separately as the averages of the regions belonging to the inner, the outer, and the central rings. In all the comparisons, individuals with subclinical renal damage had significantly lower ChT than those with normal renal function (*p* < 0.05).

**Table 1 nutrients-14-02200-t001:** Retinal arteriole wall thickness, WLR, and wall cross-sectional area in patients with MA as compared with controls. Mean differences are adjusted for BMI (see text for details).

Retinal Vessel Measures	MA Patients (*n* = 18)	Healthy Controls (*n* = 17)	*p*-Value
WLR * (mean, SD)	0.29 ± 0.05	0.25 ± 0.03	0.008
WCSA ** (mean, SD)	4876 ± 976	4004 ± 872	0.012
Internal diameter (mean, SD)	96.25 ± 13.14	94.76 ± 13.15	NS
External diameter (mean, SD)	124.78 ± 14.54	117.8 ± 15.30	NS
Wall thickness (mean, SD)	13.92 ± 1.53	11.87 ± 1.50	0.001

* WLR: wall to lumen ratio. ** WCSA: wall cross-sectional area.

## Data Availability

Not applicable.

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
