# Peer review of "Arterial Hypertension and the Hidden Disease of the Eye: Diagnostic Tools and Therapeutic Strategies"

_nutrients, 2022, doi:10.3390/nu14112200_

Round 1
Reviewer 1 Report
The authors wrote in interesting article.
Few improvements are needed.
1) please improve introduction, especially this section:In addition, vascular re- modeling in the presence of hypertension is a highly generalized phenomenon [11]. It was therefore proposed that the easily accessible vessels of the eye may be considered, to some extent, a window to the heart [8], kidney [12], and even to the brain [13] in the hyperten- sive patient.
2) please improve the academic English, there are few errors in the text
3) please evaluate to add a small sentence/section about the possible role of the microbiome. There is mounting evidence to suggest that the gut microbiome plays an important role in the development and pathogenesis of hypertension. PMID: 31276030
4) please improve discussion
Author Response
Point-by-point response
Reviewer 1
The authors wrote an interesting article. Few improvements are needed.
Response: We wish to thank the Reviewer for appreciating our work.
1) please improve introduction, especially this section: In addition, vascular re- modeling in the presence of hypertension is a highly generalized phenomenon [11]. It was therefore proposed that the easily accessible vessels of the eye may be considered, to some extent, a window to the heart [8], kidney [12], and even to the brain [13] in the hypertensive patient.
Response: we have modified the manuscript to incorporate the suggestion, thank You.
2) please improve the academic English, there are few errors in the text
Response: we have modified the manuscript to incorporate the suggestion, thank You.
3) please evaluate to add a small sentence/section about the possible role of the microbiome. There is mounting evidence to suggest that the gut microbiome plays an important role in the development and pathogenesis of hypertension.
Response: we have modified the manuscript to incorporate the suggestion, thank You.
4) please improve discussion
Response: we have modified the manuscript to incorporate the suggestion, thank You.

Reviewer 2 Report
Thank you for the opportunity to review this manuscript entitled ‘Arterial Hypertension and the Hidden Disease of the Eye: Diagnostic Tools and Therapeutic Strategies ’ by R. Del Pinto and co-authors.
In the present study, authors summarized current data of the main ocular imaging techniques in the presence of hypertension that are currently relevant from a clinical and/or research standpoint. This problem is very relevant due to the rate of increase in the incidence of hypertension is a major cardiovascular risk factor and a leading cause of morbidity and mortality world wide in. The manuscript is well written and logical and presents review of current scientific evidence appropriate for publication in Nutrients journal. However there are some key points that require clarification and adjustments, which do not allow the publication of the manuscript in its present form.
What do you understand by the term ‘Hidden Disease of the Eye‘? It is used only in the title and is not explained in the text.
In your manuscript there is no data on the study of the eye fundus in age-related macular degeneration. Why? AMD and hypertension are age-related diseases and develop at the same time. It would be logical to include the AMD in your study.
Your study lacks data on the effect of dietary calorie restriction on hypertension and fundus health. Given the title of the special issue in which you plan to publish your work, I recommend adding these data to the manuscript.
For the same reason, I recommend expanding the description of the effect of certain nutritional supplements, for example, vitamin E, resveratrol and carotenoids, in paragraph 6 of the manuscript.
Author Response
Reviewer 2
Thank you for the opportunity to review this manuscript entitled ‘Arterial Hypertension and the Hidden Disease of the Eye: Diagnostic Tools and Therapeutic Strategies ’ by R. Del Pinto and co-authors.
In the present study, authors summarized current data of the main ocular imaging techniques in the presence of hypertension that are currently relevant from a clinical and/or research standpoint. This problem is very relevant due to the rate of increase in the incidence of hypertension is a major cardiovascular risk factor and a leading cause of morbidity and mortality world wide in. The manuscript is well written and logical and presents review of current scientific evidence appropriate for publication in Nutrients journal. However there are some key points that require clarification and adjustments, which do not allow the publication of the manuscript in its present form.
Response: Thank You for appreciating this work and for the constructive criticisms and suggestions.
What do you understand by the term ‘Hidden Disease of the Eye‘? It is used only in the title and is not explained in the text.
Response: The term refers to hypertension-mediated retinal damage, a condition requiring specific instrumental assessment in the context of an organ, the eye, that can be otherwise approached by clinical examination. We have included an explanation for the title in the revised text, thank You.
In your manuscript there is no data on the study of the eye fundus in age-related macular degeneration. Why? AMD and hypertension are age-related diseases and develop at the same time. It would be logical to include the AMD in your study.
Response: we have modified the manuscript to incorporate the suggestion, thank You.
Your study lacks data on the effect of dietary calorie restriction on hypertension and fundus health. Given the title of the special issue in which you plan to publish your work, I recommend adding these data to the manuscript.
Response: we have modified the manuscript to incorporate the suggestion, thank You.
For the same reason, I recommend expanding the description of the effect of certain nutritional supplements, for example, vitamin E, resveratrol and carotenoids, in paragraph 6 of the manuscript.
Response: we have modified the manuscript to incorporate the suggestion, thank You.

Round 2
Reviewer 2 Report
I have read the changes made by the authors. I recommend accepting the manuscript for publication in its present form